# Tear Cytokine Levels in Sicca Syndrome-Related Dry Eye: A Meta-Analysis

**DOI:** 10.3390/diagnostics13132184

**Published:** 2023-06-27

**Authors:** Suad Aljohani, Ahoud Jazzar

**Affiliations:** Department of Oral Diagnostic Sciences, Faculty of Dentistry, King Abdulaziz University, Jeddah 22252, Saudi Arabia

**Keywords:** Sicca syndrome, Sjögren’s syndrome (SS), cytokines, tears, biomarkers

## Abstract

Sjogren’s syndrome (SS) is an autoimmune disease that affects exocrine glands, mainly salivary and lacrimal glands. Several studies have investigated cytokine profiles in tears in order to understand the pathogenesis of SS and find additional diagnostic markers. This systematic review and meta-analysis aimed to analyze cytokines in tears of SS patients. A systematic literature search of the Cochrane, Medline via PubMed, Scopus, and Web of Science databases was conducted using key terms related to “Sjögren’s syndrome” and “tears” combined with “biomarker”, “cytokines”, “interleukin”, and “chemokines”, following PRISMA guidelines. Article selection was subjected to certain eligibility criteria. A total of 17 articles (from 1998 and 2022) were selected for the quantitative and qualitative analysis. When compared to controls, concentrations of IFN-γ, TNF-α, IL-1α, IL-1 Ra, IL-4, IL-6, IL-8, IL-10, IL-17, IL-21 and IL-22 were consistently higher; however, IL-23 was significantly lower in patients with SS compared to the controls. Tear levels of some cytokines were significantly elevated among SS groups compared to control groups. Therefore, these cytokines could be potential biomarkers of SS. However, standardization of sample collection and analytical methods is necessary in order to translate these findings into clinical practice.

## 1. Introduction

Sicca syndrome, or Sjögren’s syndrome (SS), is a chronic autoimmune disease characterized by lymphocytic inflammatory infiltration of the exocrine gland, primarily the salivary and lacrimal glands. It contributes to defects in their activity and causes the main manifestations of SS, which are dry mouth and dry eyes. Ocular symptoms include dry or red eyes, constant ocular irritation, foreign body sensation, photophobia, pain, blurred vision, and even blindness [1].

More serious extra-glandular ocular manifestations, including B-cell non-Hodgkin’s lymphoma, have high morbidity and mortality rates. The exact causes are unknown, but a number of potential causes have been proposed. It is mediated by elevated levels of chemokines or proinflammatory cytokines that promote the recruitment and differentiation of these lymphocytes, which contribute to gland inflammation, resulting in the production of autoantibodies and the formation of germinal centers [2].

Because it affects the glands directly, SS is expected to influence the composition of the lachrymal fluid; tears thus offer tremendous potential as a non-invasive vital source containing valuable biomarkers that may serve as diagnostic indicators of local and systemic diseases. Multiple techniques are available for their collection and can affect the composition of acquired samples. Advances in analytical technologies have allowed them to detect proteins (cytokines) of low abundance in such samples. Recent studies identified multiple elevated tear chemokines and cytokines that were highly correlated with clinical parameters and disease severity [3,4].

However, until now, there has not been a comprehensive study that quantitatively synthesizes the differences between the cytokine profiles of SS and controls. Thus, the purpose of this meta-analysis was to conduct a systematic, quantitative evaluation of the available data on cytokines in SS and controls.

## 2. Methods

The Preferred Reporting Items for Systematic reviews and Meta-Analyses (PRISMA) checklist and the Cochrane Handbook of Systematic review and Meta-analysis were followed during the conduction of this study [5]. This search was registered at PROSPERO (registration number: CRD42022343160).

### 2.1. Inclusion and Exclusion Criteria

We included the studies that met the following criteria: English language publication, studies that reported the level of cytokines in the tear samples of included participants, and studies that enrolled confirmed cases of SS based on these classification criteria: the American-European Consensus Group (AECG) criteria [6], American College of Rheumatology/European League Against Rheumatism classification criteria for primary Sjögren’s syndrome (ACR/EULAR) [7] or others. Studies with five cases or more were included in order to show a statistical difference between the detected cytokines in the different groups. Other groups included healthy controls or any other disease-control subjects. Studies that included only patients with other causes of dry eye or patients with an unconfirmed diagnosis of SS, studies with less than five cases, and animal studies were excluded.

### 2.2. Search Strategy

A literature search was conducted on the 9th of March 2022 without date constriction. The following databases were searched: Cochrane, Medline via PubMed, Scopus, and Web of Science, using key terms related to “Sjögren’s syndrome” and “tears.” Several relevant keywords, including “biomarker”, “cytokines”, “interleukin”, and “chemokines”, were used in different groupings for the manual search. To enhance the sensitivity of the search strategy, the reference lists of the retrieved articles were hand-searched (records identified through other sources).

### 2.3. Study Selection

After matching the title and abstract, any possibly eligible candidate articles were screened. Then, the article was fully reviewed to determine if it fulfilled the inclusion criteria. Some articles were excluded after a review of the abstract or the full text if it was unrelated to the question’s topic. Based on the inclusion and exclusion criteria, the authors (SA and AJ) individually searched and reviewed the articles and selected references. Two investigators conducted independent critical appraisals to assess validity.

### 2.4. Data Extraction

The authors independently retrieved the following data from the eligible articles using an online data extraction form that included study design, authors, year of publication, sample size, percentage of females in each group (when applicable), type of sample being used, sampling method and cytokine quantification method, measured cytokines and main outcomes. Moreover, the mean and standard deviation (SD) of the studied cytokines in each study group were extracted.

### 2.5. Risk of Bias and Quality Assessment

Quality assessments were completed independently by the two authors (SA and AJ). Discrepancies in the assessment were resolved through discussion until a consensus was reached. The Newcastle–Ottawa Scale (NOS) was used to assess the risk of bias in the included studies. The NOS has three domains and eight elements, with a maximum score of nine. Studies with scores of 7–9 are of good quality, 4–6 are of average quality, and 0–3 are of low quality [8].

### 2.6. Data Analysis

Review Manager 5.4.1 was used to conduct all analyses. Using a fixed or random effects model, the 95% confidence interval (CI) for the standardized mean difference (SMD) between SS patients and healthy controls was estimated. Inconsistency and heterogeneity across studies were estimated using the I^2^ statistic, with values of 25%, 50%, and 75% representing low, moderate, and high levels of inconsistency, respectively. When the degree of heterogeneity was high and significant, the random-effect model was used; otherwise, the fixed-effect model was preferred. Since there were fewer than 10 studies in each comparison, thorough evaluation of publication bias was not possible.

## 3. Results

### 3.1. Search and Selection Process

The database searches found 358 studies. Four citations were retrieved from (additional/other) sources. The total number of publications found was thus 362. After removing duplicates, the number of articles was 211. Those publications were screened by title and abstract and 181 were excluded, leaving 30, which were screened in full text. After that, 13 full-text studies were excluded. Seventeen studies had the required data for qualitative and quantitative analysis (Figure 1).

### 3.2. Characteristics of Included Studies

The publication date of included studies ranged between 1998 and 2022. The number of patients with SS was 318, 297 with non-SS dry eye, and 308 were healthy controls. Tear samples were used in all of the included studies; additional samples were collected from serum in three studies, saliva in two studies, conjunctival biopsy in three studies, and conjunctival impression lysates in three studies. The American-European Consensus Criteria for Sjögren’s Syndrome (AECG) [6] was used in eight studies, the American College of Rheumatology (ACR) [9] was used in four studies, the European league against rheumatism classification criteria (EULAR) [7] was used in two studies, Lemp 1995 criteria [10] was used in two studies, and Fox et al., 1986 criteria [11] was used in one study. Regarding sampling methods, the most commonly used method was a capillary glass tube or micropipette. ELISA was used in ten studies, whereas five studies used Luminex assay, and two studies used PCR (Table 1).

### 3.3. Quality Assessments and Risk of Bias

The quality scores of the studies included in the analysis ranged from 5 to 9, reflecting variable methodological rigor across studies. The highest score of 9 was attained by two studies, namely Chung et al. [17] and Chen et al. [26], indicating their excellent methodological quality and low risk of bias. Conversely, the study by Liu et al. [22], had the lowest score of 5, pointing to potential methodological limitations. Most studies achieved a high-quality score of 8, demonstrating adequate case definition, representative cases, proper selection and definition of controls, good comparability of cases and controls, and appropriate ascertainment of exposure. A minority of studies scored 7 due to minor deficiencies in the selection of controls or comparability based on design or analysis (Table 2).

### 3.4. Meta-Analysis

#### 3.4.1. IFN-γ

The pooled analysis of six studies demonstrated that both groups had comparable IFN-γ concentrations in the patients’ tears (SMD = 0. 51, 95% CI: −0. 39 to 1. 4, *p* = 0. 27; Figure 2a). For this outcome, heterogeneity was high (I^2^ = 91.67%, *p* < 0.00001). The heterogeneity could be resolved by excluding Chen et al., 2019, Akpek et al., 2020, and Peng et al., 2021 [26,27,28]. After resolving the heterogeneity, a significant elevation in the concentration of IFN-γ was observed in the SS group compared to the control group (SMD = 0.57, 95% CI: 0.23 to 0.91, *p* = 0.001).

#### 3.4.2. IL-10

We identified four studies that reported relevant data for this outcome, involving a total of 185 participants. We did not find evidence of a clear difference between the two groups in this comparison (SMD = 0.2, 95% CI: −0.26 to 0.66, *p* = 0.39; Figure 2b). Heterogeneity within this analysis was moderate (I^2^ = 58.53%, *p* = 0.06). After excluding Akpek et al., 2020 [27], the heterogeneity was resolved (I^2^ = 0%, *p* = 0.55), and the effect size was significant (SMD = 0.40, 95% CI: 0.06 to 0.73, *p* = 0.02).

#### 3.4.3. IL-12P70

The pooled effect size of three studies showed that the concentration of IL-12P70 was comparable in both groups (SMD = 1.22, 95% CI: −0.75 to 3.2, *p* = 0.22; Figure 2c). We observed high heterogeneity in this analysis (I^2^ = 95.37%, *p* < 0.00001), which could not be resolved with sensitivity analysis.

#### 3.4.4. IL-17

The random-effect estimate of seven studies showed a significant elevation in the concentration of IL-17 in the SS compared to the control group (SMD = 1.33, 95% CI: 0.39 to 2.28, *p* = 0.006; Figure 2d). The heterogeneity was found to be high (I^2^ = 92.25%, *p* < 0.000001), which could not be resolved with sensitivity analysis.

#### 3.4.5. IL-1α

We identified three studies that reported relevant data for this outcome, involving a total of 85 participants. Patients with SS had a non-statistically significant higher IL-1α concentration in their tears compared to controls (SMD = 1.94, 95% CI: −0.02 to 3.89, *p* = 0.05; Figure 2e). This outcome had high levels of heterogeneity (I^2^ = 92%, *p* < 0.00001). We excluded Solomon et al., 2001 [14] to resolve the heterogeneity (I^2^ = 0%, *p* = 0.50), and the effect size was significant (SMD = 0.62, 95% CI: 0.09 to 1.15, *p* = 0.02).

#### 3.4.6. IL-1β

Four studies provided adequate data for this outcome, involving 151 participants. Outcomes for this subgroup had high heterogeneity. Both groups had a comparable concentration of IL-1β (SMD = 0.89, 95% CI: −0.12 to 1.9, *p* = 0.09; Figure 2f); however, the pooled data were heterogeneous (I^2^ = 87.19%, *p* < 0.00001). After excluding Solomon et al., 2001 [14], the heterogeneity was resolved, but the effect size remained non-significant (SMD = 0.28, 95% CI: −0.18 to 0.73, *p* = 0.24).

#### 3.4.7. IL-1 Ra

The fixed-effect estimate of two studies demonstrated that the concentration of IL-1 Ra was substantially higher in the SS group compared to the control group (SMD = 2.53, 95% CI: 1.88 to 3.18, *p* < 0.00001; Figure 2g). The pooled data were homogenous (I^2^ = 0%, *p* = 0.55).

#### 3.4.8. IL-2

We identified three studies relevant to this outcome, involving a total of 139 participants. There was no clear difference between SS and control (SMD = 0.26, 95% CI: −1.64 to 2.15, *p* = 0.79; Figure 3a). This outcome had high levels of heterogeneity (I^2^ = 95.95%, *p* < 0.00001), which could not be resolved with sensitivity analysis.

#### 3.4.9. IL-21

The fixed-effect estimate of two studies demonstrated a significant elevation in the tears level of IL-21 in the SS group compared to the control group (SMD = 3.67, 95% CI: 3 to 4.34, *p* < 0.00001; Figure 3b). The pooled data were homogenous (I^2^ = 0%, *p* = 0.95).

#### 3.4.10. IL-22

The pooled data of two studies showed that SS patients had a significantly higher concentration of IL-22 in their tears compared to the control group (SMD = 9.87, 95% CI: 6.14 to 13.61, *p* < 0.00001; Figure 3c). The pooled data were heterogenous (I^2^ = 72%, *p* = 0.06), which could not be resolved with sensitivity analysis.

#### 3.4.11. IL-23

The random-effect model of two studies showed that patients with SS were associated with significantly lower levels of IL-23 (SMD = −1.51, 95% CI: −2.7 to −0.32, *p* = 0.01; Figure 3d). The pooled data were heterogenous (I^2^ = 80.4%, *p* = 0.02), which could not be resolved with sensitivity analysis.

#### 3.4.12. IL-4

Four studies provided relevant data for this outcome, involving a total of 199 participants. Both groups had a comparable concentration of IL-4 (SMD = 1.48, 95% CI: −0.54 to 3.51, *p* = 0.15; Figure 3e); however, a high level of heterogeneity was detected (I^2^ = 96.91%, *p* < 0.00001). After excluding Lee et al. 2013, and Peng et al., 2021 [19,28], the pooled effect size showed a significant elevation in the IL-4 levels in the SS group compared to the control group (SMD = 3.22, 95% CI: 2.63 to 3.82, *p* < 0.00001), with no heterogeneity (I^2^ = 0%, *p* = 0.70).

#### 3.4.13. IL-6

Eight studies provided relevant data for this outcome, involving 270 participants. The random-effect model showed that patients with SS had a significantly higher level of IL-6 in their tears compared to controls (SMD = 2.15, 95% CI: 1.6 to 3.23, *p* = 0.0001; Figure 3f). Due to the high heterogeneity (I^2^ = 91%, *p* < 0.00001), we excluded, Tishler et al., 1998, Yoon et al., 2007 and Liu et al., 2017 [12,15,22], which resolved the heterogeneity (I^2^ = 47%, *p* = 0.11), and the effect size remained significant (SMD = 1.00, 95% CI: 0.58 to 1.43, *p* < 0.00001).

#### 3.4.14. IL-8

Five studies were relevant to this outcome, with a total of 195 participants. For this outcome, evidence that SS patients had a significantly higher level of IL-8 compared with control was detected (SMD = 2.02, 95% CI: 0.75 to 3.30, *p* = 0.002; Figure 3g). Heterogeneity was found to be high (I^2^ = 92%, *p* < 0.00001). By excluding Villani et al., 2013 and Akpek et al., 2020, the heterogeneity was resolved (I2 = 45%, *p* = 0.16), and the effect size remained significant (SMD = 3.01, 95% CI: 2.29 to 3.73, *p* < 0.00001) [18,27].

#### 3.4.15. TNF-α

For this outcome, six relevant studies were identified involving a total of 216 participants. The pooled analysis showed a significant elevation in the levels of TNF-α in patients with SS compared with the control group (SMD = 0.91, 95% CI: 0.16 to 1.66, *p* = 0.02; Figure 4). The heterogeneity was high (I^2^ = 84%, *p* < 0.00001); however, it could be resolved by excluding Villani et al., 2013, Akpek et al., 2020, and Peng et al., 2021 (I2 = 48%, *p* = 0.15), and the effect size remained significant (SMD = 1.20, 95% CI: 0.58 to 1.81, *p* = 0.0002) [18,27,28].

### 3.5. Results of Individual Studies

According to Yoon et al., the tear levels of CXCL9, CXCL10, and CXCL11 were significantly (*p* < 0.05) higher in the SS group in comparison to HCT and DED groups [16]. Similarly, Hernández-Ruiz et al. reported a significant elevation in the tear levels of CXCL17 in the SS group compared to the HCT (*p* < 0.0001) [23]. CCL2, CXCL8, and CXCL10 were also elevated in the tears of SS patients compared to HCT, according to Hernández-Molina et al. [29]. In terms of the precursor of IL-1b, Solomon et al. showed a significant reduction in the SS group compared to HCT [14] Chung et al., 2012 compared patients with mild and severe SS in terms of the tear level of IL-17. Their findings showed that patients with severe SS were associated with a significantly increased level of IL-17 compared to mild cases (*p* = 0.003) [17].

## 4. Discussion

Cytokines play a central role in the pathogenesis of SS in various ways, such as initiation and progression of inflammatory damage in the secretory organs. They have a direct effect on saliva and tear-producing cells leading to impaired fluid secretion and chronically stimulating B and T cell infiltration leading to lymphomagenesis [1]. A thorough review of the SS literature revealed a variety of studies that suggested various cytokines as potential biomarkers for the occurrence and course of the disease; cytokines were regularly evaluated in blood, saliva, tears, and salivary gland biopsies [30]. In murine studies, some cytokines were reported to inhibit MUC5AC secretion in goblet cells [2,31]. Thereby, increased expression of inflammatory cytokines in the conjunctiva in SS animal models and markedly decreased tear MUC5AC were observed, aggravating eye dryness [2]. Inflammatory cytokines, such as TNF-α, IFN-γ and IL-6, may be involved in goblet cell apoptosis through a diversity of signaling pathways which ultimately correlates with dry eyes [2]. To avoid the effect of long-term high concentrations of inflammatory factors, early intervention in these patients is of great importance, and using non-invasive methods of measuring cytokines, such as tear samples, is suggested.

Our findings demonstrated significantly higher levels of inflammatory mediators in tears, including IFN-γ, TNF-α, IL-1α, IL-1 Ra, IL-4, IL-6, IL-8, IL-10, IL-17, IL-21 and IL-22 in the tears of SS patients as compared to control subjects. Conversely, only IL-23 was found to be significantly lower in patients with SS compared to the control group. IL-1β, IL-2, and IL-12p70 were found to be comparable in both groups. Large means were reported in the majority of investigations, indicating significant unexplained inter-individual variance in cytokine concentrations. However, possible reasons are mentioned below. These findings generally align with the evidence that SS is associated with the production of cytokines in tears, indicating a panel suggestive of the inflammatory response. These findings shed light on the possibility of generating diagnostic or prognostic biomarkers based on these cytokines. Despite the high heterogeneity among the included studies, sensitivity analysis was able to solve this heterogeneity in the majority of the analyses.

The main sources of heterogeneity are the differences in classification criteria, sampling method, analytical method, and possible different ethnicity in the included studies. It has been established in previous articles that the levels of cytokines measured may vary depending on the sampling technique used [32,33]. Moreover, variability in the analytical techniques may have contributed substantially to heterogeneity. Each approach has its own merits and drawbacks. In spite of its sensitivity, conventional ELISA has limited use in daily clinical practice. Even though multiple ELISAs are the gold standard for measuring cytokines in a tear sample, MULTIPLEX protein analysis, a less sensitive approach, allows for simultaneous assessment of a panel of cytokines with a smaller tear volume sample [34].

Many factors might affect the accuracy of cytokine measurements; addressing the following concerns may improve data quality and facilitate future studies on SS patients’ tear fluid. The time of sampling is important since cytokine secretion has a circadian rhythm. The influence of medications on cytokine levels should be addressed as well. In addition, the diagnostic accuracy and reproducibility of the applied tests should be investigated to standardize the analytical method [35,36].

In the reported studies, Peng and his colleagues used a different method for classifying SS relying only on the serology, tear breakup time, and Schirmer test, and that may be a possible reason for the difference observed between their groups regarding IFN-γ, TNF-α and IL-4 levels [28]. According to Akpek et al., their results should be interpreted with caution because their control group might have undiagnosed underlying SS, and this might explain the higher concentrations observed in the control group for some of the analytes, such as IFN-γ, TNF-α, IL8 and IL-10 [27]. However, most of the studies did not fully confirm the absence of SS elements from their control groups except for Peng et al., where healthy controls were negative SSA and SSB, in addition to negative ocular tests [28]. Chen et al. was the only group in the category of studies reporting IFN-γ levels. They used tear strips for collection, and this might have influenced the high difference observed in their IFN-γ results [26]. Solomon et al. had a reasonably smaller sample size (*n* = 9) in comparison to other studies, but a possible explanation for the difference in their readings of IL-1 α and β could be due to age; tear fluid samples were obtained from the control group with a significantly lower age than SS [14]. Lee et al. collected a sample after instilling saline into the eye, and this might have contributed to the different IL-4 readings detected [19]. Tishler et al., Liu et al., and Yoon et al. included a smaller sample size, which might have contributed to the heterogenicity observed regarding IL-6 results [12,15,22]. In addition, one might speculate that elevated readings of IL-6 could be attributed to the limited volume of tears which is directly related to the nature of the disease. Moreover, Villani and his colleagues conducted the sample collection following the confocal examination, which might be attributed to the overproduction of observed analytes (IL-8 and TNF-α) [18].

Generally, the involved studies had relatively small cohort sizes with a maximum number of SS samples of 30 in one study [21], while the minimum included study samples was eight [15], which might affect the power of the statistical tests, thus limiting the significance of presented cytokine values. Regarding the classification criteria, there were five studies that did not follow the known classification criteria (ACR or AECG criteria). Two of them were performed before the establishment of the criteria [12,14], while the others followed different inclusion criteria, as mentioned in Table 1.

Additionally, two studies [17,18] included SS patients without matching healthy controls, but used diseased controls as stratified SS or another autoimmune disease. On the other hand, a study reported age differences between SS and healthy control groups [14]. Furthermore, it is worth mentioning that these studies might not represent the whole extent of worldwide SS characteristics. The fact that a study by Liu et al. considered not dividing the SS group according to the severity of the disease as a drawback is worth stating and should be said for most of the included studies [22]. A study by Akpek and his colleagues questioned their control group, as mentioned before [27]. Methodological limits were that the work was carried out using multiplex assays in most of the studies, which are generally not considered sensitive tools. At the same time, the ELISA that was used in the rest of the studies was intended for research only. Another methodological limit was mentioned before regarding the method of collection of tear samples.

Some limitations should be noted in our study. One of the limitations of our analysis is the relatively small number of studies available for each comparison. Specifically, we had fewer than ten studies for each contrast, which limited our ability to conduct a comprehensive evaluation of potential publication bias. Publication bias, the phenomenon where studies with positive results are more likely to be published and cited than those with negative or null results, could have potentially skewed our results. Typically, tools such as funnel plots and Egger’s regression tests are used to visually inspect and statistically evaluate publication bias. However, these tests are generally considered less reliable or meaningful when there are fewer than 10 studies. Consequently, the potential for publication bias in our meta-analysis should be acknowledged, and our results interpreted with this caveat in mind. Moreover, the significant heterogeneity in some analyses is another limitation; however, we tried to solve it with sensitivity analysis, which was effective in some scenarios. We could not assess the association between the studied biomarkers and the severity of the disease due to the lack of relevant data. In addition, we could not perform a subgroup analysis based on the assessment and analytical methods due to the scarcity of the data.

## 5. Conclusions

As a non-invasive alternative, several cytokines have promising potential as diagnostic biomarkers for SS. Standardization of sample collection and analytical techniques are needed to translate these biomarkers for use in clinical practice. Further large-scale, well-designed studies with a wide range of SS subgroups and sex/age-matched healthy controls are essential to correlate these biomarkers with the severity of the disease as a prelude for developing diagnostic and prognostic criteria based on these biomarkers.

## Figures and Tables

**Figure 1 diagnostics-13-02184-f001:**
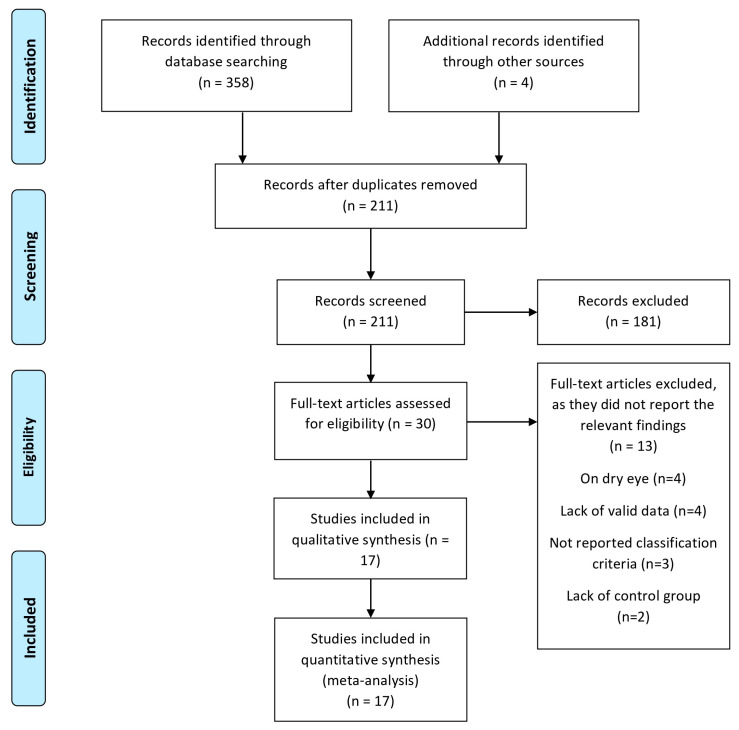
PRISMA flow diagram.

**Figure 2 diagnostics-13-02184-f002:**
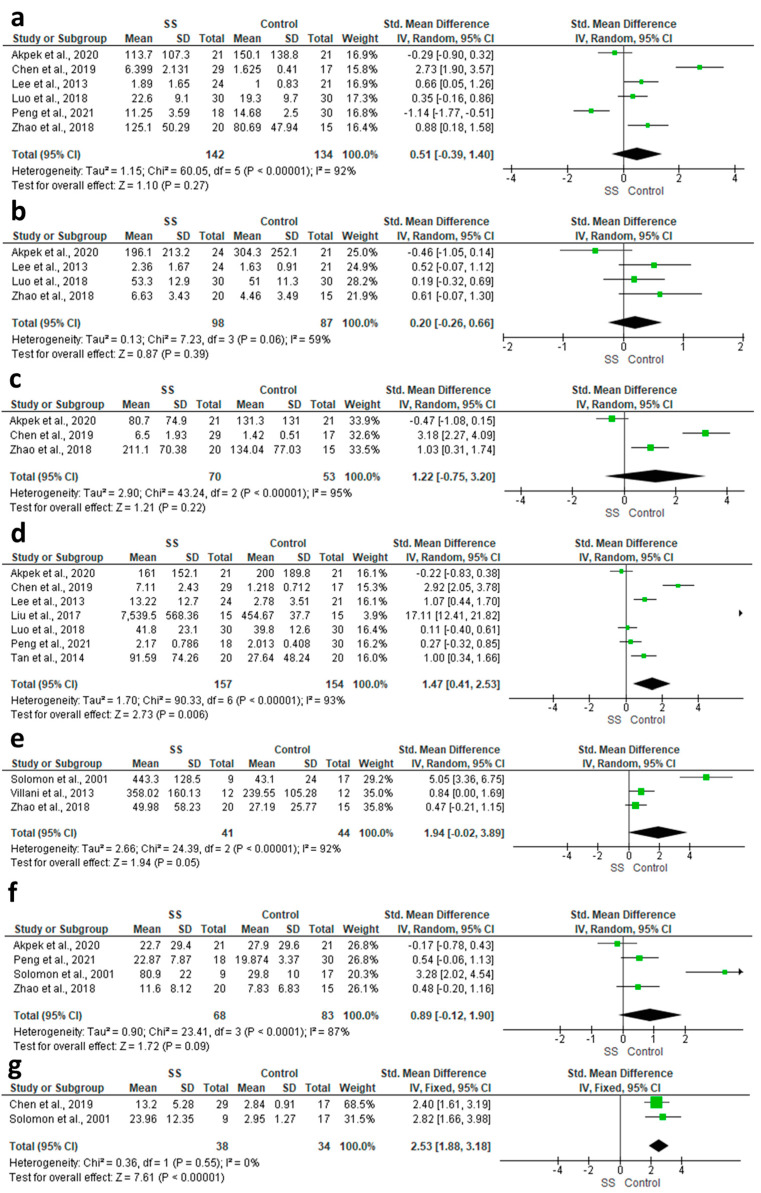
Forest plots of the difference between patients and controls in terms of (**a**) IFN-γ; (**b**) IL-10; (**c**) IL-12P70; (**d**) IL-17; (**e**) IL-1a; (**f**) IL-1β; and (**g**) IL-1 Ra.

**Figure 3 diagnostics-13-02184-f003:**
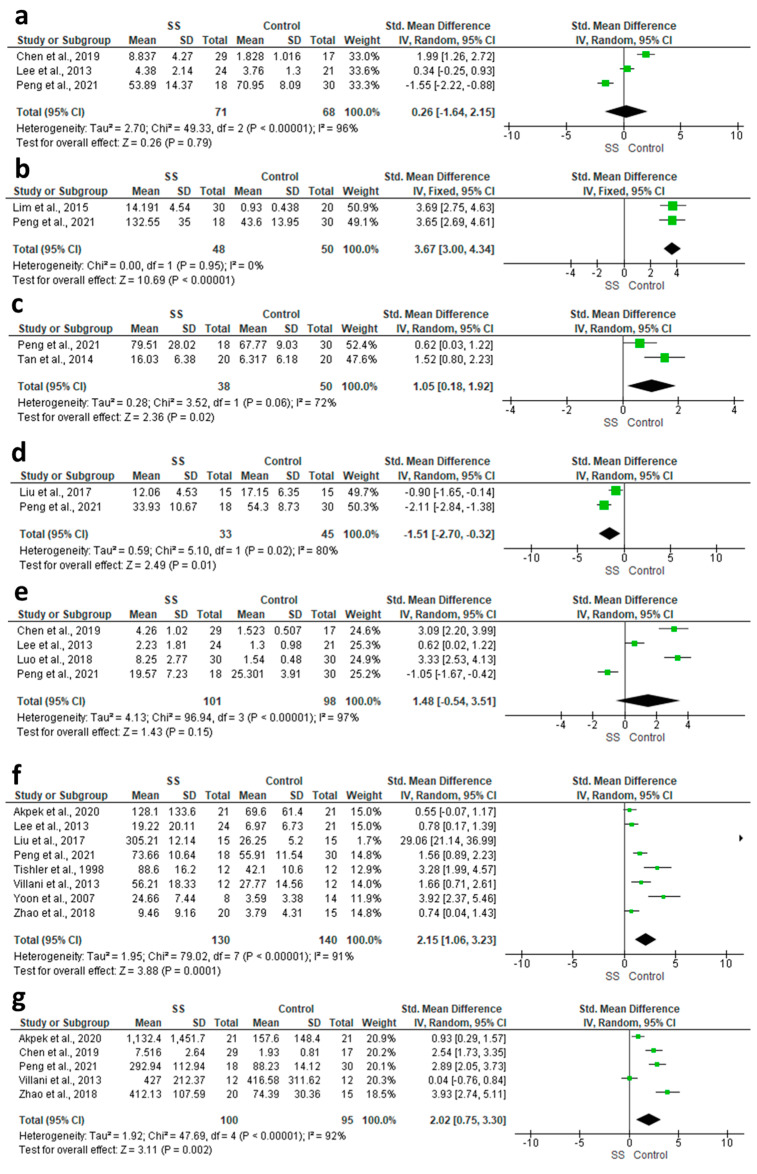
Forest plots of the difference between patients and controls in terms of (**a**) IL-2; (**b**) IL-21; (**c**) IL-22; (**d**) IL-23; (**e**) IL-4; (**f**) IL-6; and (**g**) IL-8.

**Figure 4 diagnostics-13-02184-f004:**
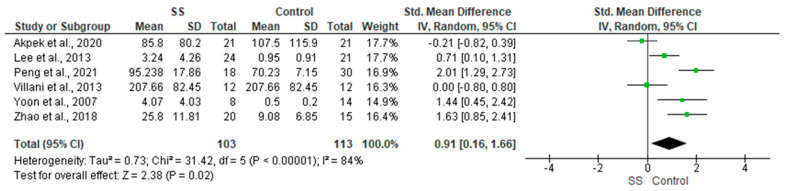
Forest plots of the difference between patients and controls in terms of TNF-α.

**Table 1 diagnostics-13-02184-t001:** Summary of included studies.

Study ID	Study Design	Study Groups	Age	Female	Type of Sample	Classification Criteria	Sampling Method	Analytical Method	Assessed Biomarkers	Findings
Tishler et al., 1998 [12]	Case-control	SS (*n* = 12)	55.3 ± 6.3	10 (83.33%)	Tears and serum	Vitali et al., 1993 [13]	Capillary Glass Micropipette	ELISA	IL-6	IL-6 was elevated in tear samples of SS (*p* < 0.05); however, no significant change detected in serum
		HCT (*n* = 12)	60.1 ± 9.9	8 (66.67%)						
		HCT (*n* = 10)	26 ± 3.8	9 (90%)						
Solomon et al., 2001 [14]	Case-control	MGD (*n* = 13)	55 ± 16.3	7 (53.85%)	Tears and Conjunctival Biopsy	Fox et al., 1986 [11]	Polyester wick	ELISA	IL-1α, IL-1Ra and IL-1β (Precursor and mature forms)	IL-1α and mature IL-1β were elevated in SS. Precursor IL-1β was decreased in SS. IL-1Ra showed no significant change.
		SS (*n* = 9)	68 ± 9.1	8 (88.89%)						
		HCT (*n* = 17)	37.76 ± 10.13	10 (58.82%)						
Yoon et al., 2007 [15]	Case-control	SS (*n* = 8)	43.63 ± 13.61	7 (87.5%)	Tears, serum, Conjunctival Biopsy, and Conjunctival Impression	AECG [6]	Capillary glass tube	ELISA, IHC, Flow cytometry	IL-6 and TNF-α	IL-6 was higher in the SS group compared to the other groups. TNF-α levels were higher in the SS group compared to HCT but not DED.
		DED (*n* = 10)	42 ± 9.75	8 (80%)						
		HCT (*n* = 14)	40.43 ± 9.47	10 (71.42%)						
Yoon et al., 2010 [16]	Case-control	SS (*n* = 16)	49.9 ± 14.6	14 (87.5%)	Tears and Conjunctival Biopsy	AECG [6]	Capillary glass tube or micropipette	ELISA, IHC, Flow cytometry	CXCL9, CXCL10 and CXCL11	CXCL9, 10, and 11 were elevated in SS in comparison to HCT and DED.
		DED (*n* = 17)	47.3 ± 17.8	13 (76.47%)						
		HCT (*n* = 15)	43.0 ± 14.9	11 (73.33%)						
Chung et al., 2012 [17]	Case-control	Mi-KCS-pSS (*n* = 60)	51.9 ± 11.7	59 (98.33%)	Tears	AECG [6]	Tear strips	ELISA	IL-17	IL-17 in the MS-KCS group was elevated compared to Mi-KCS group for both total SS and sSS.
		MS-KCS-pSS (*n* = 46)	53.4 ± 11.8	46 (100%)						
		Mi-KCS-sSS (*n* = 5)	54.5 ± 13.5	5 (100%)						
		MS-KCS-sSS (*n* = 10)	46.8 ± 12.5	10 (100%)						
Villani et al., 2013 [18]	Case-control	RA SS (*n* = 12)	51.4 ± 6.1	12 (100%)	Tears	ACR/EULAR [7] and AECG [6]	Polyurethane absorbent mini-sponge	ELISA	IL-1α, IL-6, IL-8, and TNF-α	No difference was detected between sSS/RA and RA groups. For sSS/RA group IL-1α and IL-6 were decreased after treatment (*p* < 0.01), while IL-8 and TNF-α did not show significant changes
		RA non-SS (*n* = 12)	48.8 ± 7.4	10 (83%)						
Lee et al., 2013 [19]	Case-control	SS (*n* = 24)	55.9 ± 9.96	24 (100%)	Tears	AECG [6]	Capillary glass micropipette	Multiplex bead assay	IL-2, IL-4, IL-6, IL-10 IL-17, IFN-γ, and TNF-α	IL-17, IL-6, IL-4, and TNF-α were elevated in SS compared to DED and HCT. IL-2 was elevated in SS compared to DED. IL-10 was elevated in SS compared to HCT
		DED (*n* = 25)	55.4 ± 12.44	25 (100%)						
		HCT (*n* = 21)	52.8 ± 13.19	21 (100%)						
Tan et al., 2014 [20]	Case-control	Non-SS (*n* = 20)	50.5 ± 4.6	15 (75%)	Tears	Lemp, 1995 [10]	Capillary glass micropipette	ELISA	IL-17 and IL-22	IL-17 and IL-22 were elevated in the SS group compared to HCT and non-SS patients.
		SS (*n* = 20)	52.5 ± 8.1	11 (55%)						
		HCT (*n* = 20)	54.2 ± 7.3	6 (30%)						
Lim et al., 2015 [21]	Prospective study	SS (*n* = 30)	57.3 ± 9.1	30 (100%)	Tears and Conjunctival Impression	AECG [6]	Capillary glass micropipette	Cytometric bead array and PCR	IL-21	IL-21 levels were elevated in pSS compared to DED and HCT.
		DED (*n* = 30)	58.4 ± 10.1	30 (100%)						
		HCT (*n* = 20)	54.2 ± 12.4	20 (100%)						
Liu et al., 2017 [22]	Case-control	SS (*n* = 15)	66.75± 9.38	15 (100%)	Tears and Conjunctival Impression	AECG [6]	Capillary glass micropipette	Multiplex bead assay and PCR	IL-17, IL-6, and IL-23	IL-17 and IL-6 were elevated in the SS group compared to DED and HCT. IL-23 was elevated in the SS group compared to HCT.
		DED (*n* = 15)	64.00± 11.00	15 (100%)						
		HCT (*n* = 15)	64.37± 8.34	15 (100%)						
Hernández-Ruiz et al., 2018 [23]	Case-control	SS (*n* = 28)	51.1 ± 11.09	27 (96.4%)	Tears and Saliva	AECG [6]	Tear strips	ELISA	CXCL17	CXCL17 was elevated in the SS group. Saliva results showed similar levels
		HCT (*n* = 28)		28 (100%)						
Luo et al., 2018 [24]	Case-control	SS (*n* = 30)	53.5 ± 9.2	30 (100%)	Tears and Serum	Lemp, 1995 [10]	Capillary glass tube	ELISA	IL-33, IL-4, IL-5, IFN-γ, IL-10, IL-17, and TGF-β	IL-4 and IL-5 were elevated in the SS group compared to HCT and non-SS. IL-33 was elevated in the SS group compared to non-SS.
		Non-SS (*n* = 30)	51.6 ± 8.3	30 (100%)						
		HCT (*n* = 30)	50.6 ± 6.7	30 (100%)						
Zhao et al., 2018 [25]	Prospective study	SS (*n* = 20)	NA	NA	Tears	ACR [9]	Capillary glass tube	Luminex assay and ELISA	TNF-α, IL-1α, IL-1β, IL-6, IL-8, IL-10, IL-12P70, IL-13, IFN-γ, EGF and MIP-1α	TNF-α, IL-6, IL-8, and IL-12P70 were elevated in all patient groups compared to HCT. IL-8 and TNF-α were also elevated in SS-ATD compared to the other diseased groups
		Non-SS (*n* = 20)								
		MGD (*n* = 15)								
		HCT (*n* = 15)								
Chen et al., 2019 [26]	Case-control	SS (*n* = 29)	56.8 ± 13.0	29 (100%)	Tears and saliva	AECG [6]	Tear strips	Luminex assay	IL-1RA, IL-2, IL-4, IL-17A, IFN-γ, MIP-1b, L-8, IL-12p70, and IP-10	IL-1RA, IL-2, IL-4, IL-17A, IFN-γ, and MIP-1b were elevated in the SS group compared to DED and HCT. L-8, IL-12p70, and IP-10 were elevated in the SS group compared to HCT. Salivary levels of IP-10 and MIP-1a were elevated in the SS group compared to DED and HCT
		DED (*n* = 20)	51.7 ± 10.6	20 (100%)						
		HCT (*n* = 17)	45.4 ± 10.9	17 (100%)						
Akpek et al., 2020 [27]	Prospective study	SS (*n* = 21)	63.3 ± 10.1	16 (76.2%)	Tears	ACR [9]	Capillary glass tube	Luminex assay	TNF-α, IFN-γ, IL-1β, IL-6, IL-8, IL-10, IL-12p70, and IL-17A.	IL-8 was elevated in SS compared to HCT and DED. Among all groups IL-6, IL-12p70 and IL-8 were elevated.
		DED (*n* = 20)	62.6 ± 7.7	17 (85.0%)						
		HCT (*n* = 21)	60.2 ± 5.6	14 (66.7%)						
Peng et al., 2021 [28]	Case-control	DED (*n* = 13)	35.2 ± 11.7	10 (76.9%)	Tears	SIT <10, TBUT <10, and anti-dsDNA or anti-Smith antibodies negative	Capillary glass tube	Luminex assay	IL-1β, IL-2, IL-4, IFN-γ, IL 6, IL-8, IL-17F, TNF-α, IL-21, IL-22, and IL-23	TNF-α and IL-6 were elevated in SS compared to HCT. IL-8 and IL-21 were elevated in SS compared to HCT and DED. IL-23 was decreased in SS compared to HCT. IFN-γ, IL-2, and IL-4 were decreased in SS compared to HCT and DED.
		SLE (*n* = 17)	34.1 ± 9.3	16 (94.1%)						
		SS (*n* = 18)	41.6± 12.6	15 (83.3%)						
		HCT (*n* = 30)	33.7 ± 9.6	15 (50%)						
Hernández-Molina et al., 2022 [29]	Case-control	SS (*n* = 21)	NA	19 (90.4%)	Tears	ACR/EULAR [7]	Tear strips	Luminex assay	CCL2, CXCL8, and CXCL10	CXCL10 and CCL2 were decreased in the SS group, while CXCL8 was similar.
		HCT (*n* = 21)	NA	21 (100%)						

SS, Sjögren’s syndrome; HCT, healthy control; EGF, Epidermal Growth Factor; MGD, rosacea-associated Meibomian gland disease; SS-ATD, Sjögren’s syndrome aqueous tear deficiency; DED, Dry eye disease; AECG, The American-European Consensus Group criteria; IHC, Immunohistochemistry; breakup, Tear film break-up time; ACR/EULAR, The American College of Rheumatology/European league against rheumatism classification criteria; RA, Rheumatoid arthritis; IL, interleukin; TNF-a, tumor necrosis factor alpha; INF, Interferon.

**Table 2 diagnostics-13-02184-t002:** Quality assessments and risk of bias of observational studies.

Study ID	Selection				Comparability	Exposure			Total
	Is the Case Definition Adequate?	Representativeness of the Cases	Selection of Controls	Definition of Controls	Comparability of Cases and Controls on the Basis of the Design or Analysis	Ascertainment of Exposure	Same Method of Ascertainment for Cases and Controls	Non-Response Rate	
Tishler et al., 1998 [12]	*****	*****	*****		******	*****	*****	*****	8
Solomon et al., 2001 [14]	*****	*****	*****	*****	*****	*****	*****	*****	8
Yoon et al., 2007 [15]	*****	*****		*****	*****	*****	*****	*****	7
Yoon et al., 2010 [16]	*****	*****		*****	*****	*****	*****	*****	7
Chung et al., 2012 [17]	*****	*****	*****	*****	******	*****	*****	*****	9
Villani et al., 2013 [18]	*****	*****	*****	*****	*****	*****	*****	*****	8
Lee et al., 2013 [19]	*****	*****			******	*****	*****	*****	7
Tan et al., 2014 [20]	*****	*****	*****	*****	*****	*****	*****	*****	8
Lim et al., 2015 [21]	*****	*****			******	*****	*****	*****	7
Liu et al., 2017 [22]	*****	*****			** * **	*****	*****		5
Hernández-Ruiz et al., 2018 [23]	*****	*****	*****	*****	*****	*****	*****	*****	8
Luo et al., 2018 [24]	*****	*****			******	*****	*****	*****	7
Zhao et al., 2018 [25]	*****	*****	*****		******	*****	*****	*****	8
Chen et al., 2019 [26]	*****	*****	*****	*****	******	*****	*****	*****	9
Akpek et al., 2020 [27]	*****	*****	*****	*****	*****	*****	*****	*****	8
Peng et al., 2021 [28]	*****	*****	*****	*****	*****	*****	*****	*****	8
Hernández-Molina et al., 2022 [29]	*****	*****	*****	*****	*****	*****	*****	*****	8

## Data Availability

Data available upon request from the corresponding author.

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
