# Peer review of "Tear Cytokine Levels in Sicca Syndrome-Related Dry Eye: A Meta-Analysis"

_diagnostics, 2023, doi:10.3390/diagnostics13132184_

Round 1

Reviewer 1 Report

It is an interesting and well-organized manuscript; however, I have a few concerns that the authors should address. The authors should discuss the impact of the observed non-significance in the forest plots, as it relates to cytokines such as IL-4 and IFN-gamma. There is no discussion of IL-2 on page 10 of 15. I would like the author(s) to address these suggestions/comments raised by the reviewer.

Line 20: It appears that this is a sentence fragment, and as such, it is incomplete.

Line 45: What are these arising studies?

Lines 85 – 89: The sentence should be reworded to make it more clarity.

 Line 106: “we were unable to conduct a thorough evaluation of publication bias” Address this in the discussion.

Line 203: Provide a detailed description of table 2.

Line 207 and 210: Render “IFN” as “IFN

Figure 2 and figure 3: Provide a figure legend with a detailed description of each of the subsets of figure 2 and 3.

Figure 2d and Figure 2g: Provide an elaborate explanation of the forest plot for IL-17 and IL-1Ra.

Figure 3e: Elaborate on the weighting of the studies by Luo et al and Peng et al.

Figure 3g: Elaborate on the weighting of the studies by Peng et al. and Chen et al.

Lie 297: Render “TNF-a” as “TNF-

Line 305, 307, and 311: Include the cited reference for Yoon et al., Hernández-Ruiz et al, and Chung et al.

Line 321: Cite two or more references to support this claim.

Lines 322 – 324: The sentence should be reworded for better clarity.

Lines 324 – 326: Cite your source and provide examples of these cytokines.

Line 330 - 331: Render “higher levels of the tear inflammatory mediators” as “higher levels of inflammatory mediators in tears”

Lines 376 – 378: The sentence should be reworded for better clarity.

Lines403 – 404: Why was the sensitivity analysis not effective in resolving the heterogeneity?

Lines 410 – 411: The sentence should be reworded for better clarity.

See comments.

Author Response

It is an interesting and well-organized manuscript; however, I have a few concerns that the authors should address. The authors should discuss the impact of the observed non-significance in the forest plots, as it relates to cytokines such as IL-4 and IFN-gamma. There is no discussion of IL-2 on page 10 of 15. I would like the author(s) to address these suggestions/comments raised by the reviewer.

Line 20: It appears that this is a sentence fragment, and as such, it is incomplete.

Thank you very much for your valuable comments. This comment was addressed.

Line 45: What are these arising studies?

Another study was added.

Lines 85 – 89: The sentence should be reworded to make it more clarity.

This comment was addressed.

 Line 106: “we were unable to conduct a thorough evaluation of publication bias” Address this in the discussion.

Thank you for your comment. We added this paragraph to the limitations section “One of the limitations of our review and analysis is the relatively small number of studies available for each comparison. Specifically, we had fewer than ten studies for each contrast, which limited our ability to conduct a comprehensive evaluation of potential publication bias. Publication bias, the phenomenon where studies with positive results are more likely to be published and cited than those with negative or null results, could have potentially skewed our results. Typically, tools such as funnel plots and Egger's regression tests are used to visually inspect and statistically evaluate publication bias. However, these tests are generally considered less reliable or meaningful when there are fewer than 10 studies. Consequently, the potential for publication bias in our meta-analysis should be acknowledged, and our results interpreted with this caveat in mind."

Line 203: Provide a detailed description of table 2.

Thank you for your comments. We have added more descriptions about this table “The quality scores of the studies included in the analysis ranged from 5 to 9, reflecting variable methodological rigour across studies. The highest score of 9 was attained by two studies, namely Chung et al. [13], and Chen et al. [23], indicating their excellent methodological quality and low risk of bias. Conversely, the study by Liu et al. [26], had the lowest score of 5, pointing to potential methodological limitations. Most studies achieved a high-quality score of 8, demonstrating adequate case definition, representative cases, proper selection and definition of controls, good comparability of cases and controls, and appropriate ascertainment of exposure. A minority of studies scored 7 due to minor deficiencies in the selection of controls or comparability based on design or analysis”.

Line 207 and 210: Render “IFN” as “IFN”

 Done.

Figure 2 and figure 3: Provide a figure legend with a detailed description of each of the subsets of figure 2 and 3.

Thank you for your comment. We have added a detailed legend as requested.

Figure 2d and Figure 2g: Provide an elaborate explanation of the forest plot for IL-17 and IL-1Ra.

We already interpreted these two plots in the manuscript as follows: “The random-effect estimate of seven studies showed a significant elevation in the concentration of IL-17 in the SS compared to the control group (SMD= 1.33, 95% CI: 0.39 to 2.28, p =0.006; Figure 2d). The heterogeneity was found to be high (I² = 92.25%, p<0.000001), which could not be resolved with sensitivity analysis” and “The fixed-effect estimate of two studies demonstrated that the concentration of IL-1 Ra was substantially higher in the SS group compared to the control group (SMD= 2.53, 95% CI: 1.88 to 3.18, P<0.00001; Figure 2g). The pooled data were homogenous (I2= 0%, p=0.55).”

Figure 3e: Elaborate on the weighting of the studies by Luo et al and Peng et al.

 Figure 3g: Elaborate on the weighting of the studies by Peng et al. and Chen et al.

In these forest plots, the "weight" refers to the influence or contribution of each individual study in the pooled estimate or meta-analysis. This weight is typically determined by the size of the study (sample size) and the variance of the effect estimate. Larger studies or studies with lower variance (greater precision) are generally given more weight because their effect estimates are considered to be more reliable.

Lie 297: Render “TNF-a” as “TNF-

Done.

Line 305, 307, and 311: Include the cited reference for Yoon et al., Hernández-Ruiz et al, and Chung et al.

Included.

Line 321: Cite two or more references to support this claim.

Done.

Lines 322 – 324: The sentence should be reworded for better clarity.

Done.

Lines 324 – 326: Cite your source and provide examples of these cytokines.

Done

Line 330 - 331: Render “higher levels of the tear inflammatory mediators” as “higher levels of inflammatory mediators in tears”

Done

Lines 376 – 378: The sentence should be reworded for better clarity.

Done

Lines403 – 404: Why was the sensitivity analysis not effective in resolving the heterogeneity?

Thank you for your comment. Sensitivity analysis helps identify studies significantly contributing to heterogeneity. However, it may not fully resolve heterogeneity arising from multiple factors like variations in study design, sample size, or population characteristics. In our case, sensitivity analysis did identify some influential studies, but some heterogeneity remained unexplained. This could be due to multi-factorial reasons beyond the scope of sensitivity analysis, which is a limitation we've noted in our discussion. We appreciate your feedback and the opportunity to clarify this issue.

Lines 410 – 411: The sentence should be reworded for better clarity.

Done.

Reviewer 2 Report

I found some spelling errors and extra spaces between words, for example:

-The figure 1 has compressed boxes, probably due to automatic style correction, so need editting again.

-line 407: is sacristy, should be scarsity

-line 237: β is in bold and should be normal font

-mix of greek and latin symbols of ILs, e.g.: in line 237 is ILβ and in line 240 is IL1b - lack of unique style of writing of ILs symbols - it should be unified in one style.

-not specified why division of different ILs into two figures 2 nad 3? For readers better should be placing ILs in growing number, e.g. IL-1 alpha, IL-1beta, IL-2, IL-4 and so on. Now, the reader is confused, where to find the cytokine of interest, if it is not placed in some order easier to understand.

Author Response

I found some spelling errors and extra spaces between words, for example:

-The figure 1 has compressed boxes, probably due to automatic style correction, so need editting again.

Thank you for your valuable comment. The image has been changed to jpg format and cannot be distorted.

-line 407: is sacristy, should be scarsity

Done.

-line 237: β is in bold and should be normal font.

Done

-mix of greek and latin symbols of ILs, e.g.: in line 237 is ILβ and in line 240 is IL1b - lack of unique style of writing of ILs symbols - it should be unified in one style.

Done

-not specified why division of different ILs into two figures 2 nad 3? For readers better should be placing ILs in growing number, e.g. IL-1 alpha, IL-1beta, IL-2, IL-4 and so on. Now, the reader is confused, where to find the cytokine of interest, if it is not placed in some order easier to understand.

Dear reviewer, thank you for your comment. We have cited each figure/plot in the text of the results section so the reader can easily navigate the figure of interest by tracking the number of the figure.

Round 2

Reviewer 1 Report

I am satisfied with your response to my questions/comments.